# IL-17B protects against uropathogenic *E. coli*-induced kidney injury via macrophage infiltration modulation

Changying Wang,[1,2] Min Liu,[1] Luyue Wang,[1] Yixin Zhang,[1] Zihan Niu,[1] Xue Liu,[1] Huabao Xiong[1,2]

**ABSTRACT** Urinary tract infections (UTIs) caused by uropathogenic *Escherichia coli* (UPEC) are prevalent among women. UPEC infection can lead to kidney injury, and currently, there are no effective methods to mitigate this damage. Previous studies have shown that macrophage infiltration is closely associated with kidney injury induced by UTIs. However, the mechanisms underlying macrophage recruitment to the kidneys remain unclear. In this study, IL-17B may play a critical role in protecting against UPEC-induced kidney injury by modulating macrophage infiltration and bacterial colonization. Compared with wild-type (WT) mice, IL-17B$^{-/-}$ mice exhibited significantly higher mortality and more M1-type macrophage infiltration, which was associated with renal injury. rIL-17B treatment significantly reduced macrophage infiltration in the kidneys of CFT073-infected mice. Additionally, analysis of chemokines indicated that IL-17B is a key regulator of macrophage recruitment by influencing the expression of CCL2, CCL3, and CCL7. Overall, IL-17B can serve as a crucial cytokine for treating urinary tract infections and mitigating kidney damage.

**IMPORTANCE** Urinary tract infections (UTIs) are a prevalent bacterial infectious disease that significantly affects women due to their recurrent nature and tissue damage. The increase of M1-type macrophages in infected kidneys is related to kidney injury. Investigating the mechanisms regulating macrophage infiltration could provide novel insights and strategies for treating UTIs. Previous studies have demonstrated the crucial role of IL-17A in modulating inflammation induced by UTIs, while the function of IL-17B remains poorly understood. Our research revealed that IL-17B deficiency enhanced macrophage infiltration and significantly increased mortality rates in mice during UTIs. Recombinant IL-17B treatment markedly alleviates renal damage caused by inflammation. These findings suggest that IL-17B could serve as a potential adjunctive therapeutic approach for severe UTIs, effectively controlling excessive inflammation-induced renal injury.

**KEYWORDS** uropathogenic *Escherichia coli*, kidney injury, IL-17B, macrophages, chemokines

Urinary tract infections (UTIs) is a common bacterial disease that frequently affects human health. Uropathogenic *Escherichia coli* (UPEC) is the primary cause of UTIs, accounting for over 75% of cases (1). UTIs can involve the entire urinary system. Once bacteria enter the bladder, they can cause cystitis; if the bacteria ascend through the ureters to the kidneys, they may cause pyelonephritis. In severe cases, bacteria can enter the bloodstream, leading to systemic endotoxin shock and widespread organ damage. In acute pyelonephritis, both bacterial infection and an excessive immune response, including cytokine release and excessive immune cell infiltration, can contribute to kidney injury (2, 3). Due to physiological structure differences, UTIs are more common in women, with approximately 50% experiencing at least one episode, although about

**Peer Reviewers** Basel H. Abuaita, Louisiana State University School of Veterinary Medicine, Baton Rouge, Louisiana, USA; Parmanand Prabhakar, Bihar Animal Sciences University, Kishanganj, Bihar, India

Address correspondence to Huabao Xiong, xionghbl@163.com.

The authors declare no conflict of interest.

See the funding table on p. 12.

10% of men are also affected. Around 25% of patients with UTIs experience recurrent infections (4–7). More than 60% of recurrent infections are caused by the same UPEC strain as the initial episode, likely due to persistent reservoirs of UPEC within the host (8–10). Studies have reported that the colon serves as a major reservoir, where UPEC resists clearance by expressing FimH. This protein facilitates UPEC colonization in the colon and enables translocation to the lower urinary tract, contributing to recurrent infections (8, 10). Antibiotics are effective for treating UTIs, and early administration can prevent renal scarring caused by severe upper UTIs, especially in children. However, recurrent infections and rising antibiotic resistance have reduced the overall effectiveness of treatment, increasing the risk of tissue damage in the urinary tract (11, 12). Therefore, it is urgent for us to further explore the mechanism of kidney damage caused by UTIs and identify novel approaches for the treatment of UTIs to prevent kidney damage.

The IL-17 family is a key group of cytokines involved in the defense against bacterial infections and plays an important role in UTIs caused by UPEC, *Proteus mirabilis*, *Pseudomonas aeruginosa*, and *Candida* (13–16). During infections with *Proteus mirabilis*, *Pseudomonas aeruginosa*, *Candida*, and UPEC, the expression levels of IL-17 in tissues or cells increase significantly (14–18). Inhibition of IL-17 may worsen UTIs caused by *Candida* (18). In UPEC-induced UTIs, IL-17 plays a critical regulatory role in female mice, while it has little effect on male mice, possibly due to that male mice have a milder inflammatory response in the kidneys after infection (17). Previous studies have shown that in both acute and chronic UTIs, IL-17A influences UPEC colonization in kidney tissues. In the absence of IL-17A, UPEC colonization and the incidence of pyelonephritis in the kidneys increase markedly. In bladder tissues infected by UPEC, γδT cells and Th17 cells are major sources of IL-17A (13, 19). However, the specific roles of other IL-17 family members in UTIs remain poorly understood.

IL-17B, a recently identified member of the IL-17 family, exerts its biological effects through binding to its receptor. The IL-17B receptor (IL-17RB) can also form a heterodimer with the IL-17A receptor (IL-17RA), enabling binding to IL-25 and participating in responses to bacterial infections and anti-parasitic immunity. It is also associated with allergic and atopic diseases (20–22). Although IL-17B and IL-25 share a receptor subunit, they exhibit opposing effects in many inflammation-related diseases. For example, in models of acute colitis, airway inflammation, and infection with *Citrobacter rodentium*, IL-25 facilitates the onset and progression of these diseases, whereas IL-17B suppresses it. Thus, IL-17B deficiency can exacerbate inflammation (21). However, existing studies on IL-17B have yielded conflicting results. In arthritis models, administration of IL-17B-neutralizing antibodies significantly alleviates collagen-induced arthritis (21, 23). IL-17B also modulates tissue damage by regulating macrophage infiltration. For instance, in sciatic nerve injury, Schwann cells secrete IL-17B in an autocrine manner, activating the IL-17B/IL-17RB signaling pathway. This activation promotes the secretion of CCL2, CCL3, CCL5, CCL19, CCL22, and G-CSF by Schwann cells, facilitating macrophage infiltration and supporting nerve repair (24). Given that UTIs are a common bacterial inflammatory disease, this study aims to investigate the role of IL-17B in UTIs, particularly in kidney injury, and to explore the potential of IL-17B as a therapeutic target for preventing and treating UTI-associated kidney damage.

## MATERIALS AND METHODS

### Animals and strains

Female C57BL/6J mice, aged 6–8 weeks, were used. WT mice were obtained from Pengyue (Shandong, China). IL-17B knockout (IL-17B$^{-/-}$) mice on a C57BL/6J background were purchased from GemPharmatech (Jiangsu, China). All animals were housed in a specific-pathogen-free (SPF) facility at Jining Medical University. All experiments were conducted in accordance with institutional regulations for animal care and approved by the Animal Ethics Committee of Jining Medical University (Ethics Review Number:

JNMC-2023-DW-133). The UPEC strain CFT073 was kindly provided by Professor Quan Wang.

## Construction of the urinary tract infection model

WT mice and IL-17B$^{-/-}$ mice were randomly assigned to control and CFT073 infection groups. CFT073 was first cultured overnight on lysogeny broth (LB) agar plates, followed by overnight incubation in liquid medium under static conditions. The bacterial suspension was used to infect the mice intraurethrally. Mice were anesthetized with pentobarbital, and each mouse was administered $1 \times 10^8$ colony-forming unit (CFU) of CFT073 in 50 µL twice, with a 3 h interval. Mice in the control group received the same volume of PBS. Twenty-four hours after infection, the mice were euthanized (25), and tissue samples were collected for analysis. In the survival experiment, we used a higher dose of infection in order to observe the death of the mice. Each mouse was injected with $1 \times 10^9$ CFU of CFT073 twice, 3 h apart, and their survival was continuously monitored (26–28).

## Clodronate liposomes eliminate macrophages, and neutralizing antibodies eliminate neutrophils

200 µL of control liposomes (con-liposome) or clodronate liposomes (Clod liposomes) (F70101C-A, FormuMax) were injected intravenously to eliminate phagocytes (29). Anti-Ly6G antibody (10 mg/kg) (BE0075-1, InVivoMAb anti-mouse Ly6G, Bioxcell) was used to deplete neutrophils by intravenous administration. Isotype control antibody (BE0089, InVivoMAb rat IgG2a, Bioxcell) was used as a control (30). After 24 h, the mice were infected with CFT073.

## Recombinant IL-17B treatment

1µg / mouse recombinant IL-17B (rIL-17B) was injected intravenously, and the mice were infected with CFT073 at the same time. The saline (NS) was used as control (31).

## H&E staining

The kidneys of infected mice were collected and fixed in tissue fixative for 48 h. The tissues were then embedded in paraffin, sectioned at a thickness of 5 µm, and stained with H&E. Kidney damage was evaluated under a microscope. Pathological scoring was performed using the following criteria: if damage was confined to regions outside the renal cortex, 0: no damage; 1: mild; 2: moderate; 3: severe. If the damage reached the renal cortex, 4: mild; 5: moderate; 6: severe. This assessment is primarily based on histopathological changes at the corticomedullary junction, like cell infiltration, tubular atrophy, and interstitial inflammation (26, 32). The pathological scores of the bladders are devised according to Wang et al. (33). This standard is as follows: 0: normal; 1: focal or multifocal subepithelial inflammatory cell infiltration; 2: presence of edema with diffuse subepithelial inflammatory cell infiltration; 3: marked subepithelial inflammatory cell infiltration accompanied by neutrophil migration into the bladder mucosal epithelium with necrosis; 4: the same phenomenon as grade 3, following extension of inflammatory infiltration into the smooth muscle layer; 5: loss of surface epithelium, full-thickness inflammatory cell infiltration, and extensive necrosis.

## Bacterial colonization experiment

Kidneys of CFT073-infected mice were harvested, and 1 mL of PBS containing 0.025% Triton X-100 was added. The tissue was homogenized, serially diluted, and plated for colony counting to assess bacterial colonization.

## ELISA

Kidneys were lysed in 1 mL of buffer containing protease inhibitors (Beyotime, P1005). The supernatant was collected and analyzed using commercial ELISA kits according

to the manufacturer's instructions. ELISA kits for mouse IL-1β (432601), IL-12/23 p40 (431601), IL-6 (431301), TNFα (430901), IL-17A (432504), and IFN-γ (430804) were purchased from BioLegend, USA. The absorbance was measured at 450 nm.

## The level of creatinine

After the mice were anesthetized, peripheral blood was collected, and plasma was retained. The creatinine level was measured in accordance with the manufacturer's instructions of the reagent (Nanjing Jiancheng Bioengineering Institute, C011-2-1).

## Real-time quantitative PCR (qPCR)

The kidneys of mice infected by CFT073 after 24 h were taken and homogenized in RNAiso Plus (9109, TaKaRa) to extract total RNA. Complementary DNA (cDNA) was synthesized from 1 μg of total RNA with a reverse transcription first-strand cDNA synthesis kit (312-01, Vazyme) following the manufacturer's protocol. The synthesized cDNA was used for qRT-PCR using a SYBR Green PCR Master Mix (Q311-02) and detected with the LightCycler480II system. The β-actin was used as a housekeeping gene, and the gene expression was calculated using the $2^{-\Delta\Delta Ct}$ method for fold change. All the primer sequences were synthesized by Sangon (Shanghai, China) and listed in Table S1.

## Flow cytometry

Mouse spleens were mechanically dissociated to form single-cell suspensions. Kidneys were minced and enzymatically digested with type IV collagenase and DNase I for 40 mins. The digested tissues were filtered to obtain single-cell suspensions. Red blood cells in the spleen, kidney, and peripheral blood were lysed with red blood cell lysis buffer. The remaining cells were incubated with CD16/32 blocking for 10 min and then labeled with specific antibodies, analyzed using a BD FACSVerse flow cytometer, and interpreted using FlowJo software. The antibodies are listed in Table S2.

## Culture and infection of BMDMs

Macrophages were obtained from the bone marrow of mice (BMDMs). After mice were euthanized, the bone marrow was flushed out with PBS, and red blood cells were lysed. Subsequently, the remaining cells were cultured in Dulbecco's modified Eagle medium (DMEM) supplemented with M-CSF (10 ng/mL) for 7 days. The bacterial strain was cultured in LB at 37°C. Bacteria were harvested when the OD600 was between 0.6 and 0.8. BMDMs were seeded into 6-well tissue plates at $2 \times 10^6$ cells per well. And the cells were infected by bacteria (MOI = 0.01) for 6 h. Cells were collected for qPCR.

## Statistical analysis

Flow cytometry data were analyzed using FlowJo VX. Graphs and statistical analyses were performed using GraphPad Prism 8.0.1 and SPSS Statistics. Data were expressed as mean ± SEM, and comparisons between two groups were conducted using data from at least three independent experiments.

## RESULTS

### IL-17B protects mice and prevents kidney injury caused by UPEC infection

To investigate the role of IL-17B in UPEC infection, we first measured IL-17B and IL-17RB mRNA levels in the kidneys of mice after infection. The results showed a significant increase in both IL-17B and IL-17RB expression following infection (Fig. 1A). This suggests that IL-17B is involved in the host response to CFT073-induced UTIs. Previous studies have established that the expression of IL-17A increased in UPEC-infected bladders and kidneys. When IL-17A is deficient, the colonization of UPEC in the kidneys and bladders of mice significantly increase. This study also showed that the expression level of IL-17A in

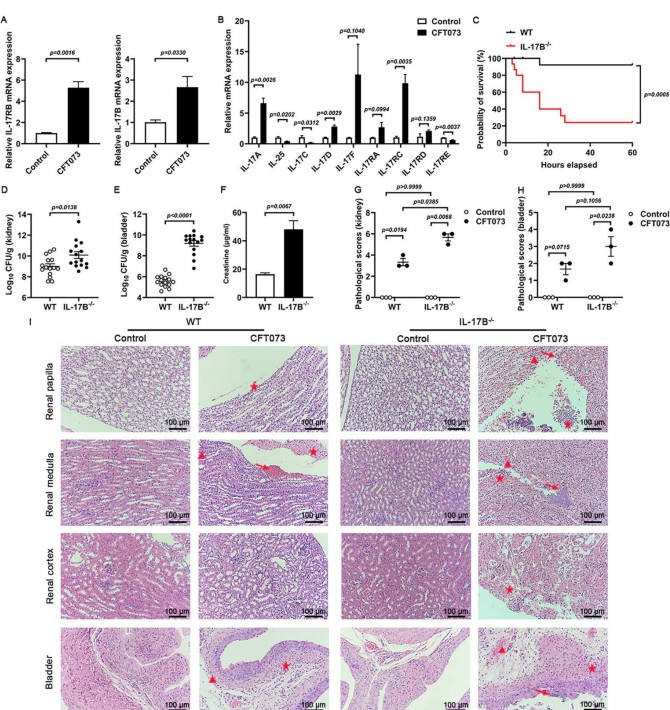

**FIG 1** The absence of IL-17B leads to aggravated kidney injury in the CFT073 infection group. (A and B) Expression levels of IL-17B, IL-17RB, and various members of the IL-17 family mRNA in the kidneys of CFT073-infected and uninfected mice ($n = 3$). (C) Survival rates of WT and IL-17B$^{-/-}$ mice infected intraurethrally with $1 \times 10^9$ CFU of CFT073, administered twice at 3 h intervals ($n = 15$). (D and E) Bacterial load in the kidney and bladder 24 h after infection with $1 \times 10^8$ CFU of CFT073 ($n = 15$). (F) The level of creatinine in the plasma of mice after infection for 24 h. (G–I) Kidney and bladder injury scores based on H&E staining, along with representative histological images. Triangles indicate hemorrhage, arrows denote tissue damage, and stars indicate inflammatory cell infiltration. 200×, 100 µm ($n = 3$). Data are presented as mean ± SEM. Statistical comparisons were performed using (A and B, D–F) $t$-test. (C) Kaplan-Meier survival curve Log-rank (Mantel-Cox) test (G and H) was performed using two-way ANOVA. Each experiment was performed in triplicate at least.

the kidneys of mice was significantly increased (Fig. 1B) (13, 19, 34, 35). At the same time, other members of the IL-17 family in the kidneys exhibit changes after being infected by CFT073 (Fig. 1B). To further examine the role of IL-17B, WT and IL-17B$^{-/-}$ mice were infected with $1 \times 10^9$ CFU of CFT073. IL-17B deficiency leads to a significantly increased mortality rate (Fig. 1C). When infected with $1 \times 10^8$ CFU of CFT073, bacterial colony counts revealed significantly higher bacterial colonization in the kidneys and bladders of IL-17B$^{-/-}$ mice compared to WT mice (Fig. 1D and E). Meanwhile, the level of creatinine in the plasma of IL-17B$^{-/-}$ mice increased significantly compared with WT mice (Fig. 1F). Histopathological analysis using H&E staining demonstrated that IL-17B$^{-/-}$ mice exhibited more extensive inflammatory cell infiltration and severe renal tissue damage, compared to WT mice upon CFT073 infection (Fig. 1G through I). These findings indicate that IL-17B deficiency enhances bacterial colonization and exacerbates renal injury during CFT073-induced urinary tract infection.

## IL-17B attenuates renal injury by influencing various immune cell infiltration

Our previous studies identified macrophages as key contributors to renal injury in UTIs, and this result has been published (36). In this study, we also examined the polarization of macrophages. Flow cytometry revealed that IL-17B$^{-/-}$ mice had a slight reduction in inflammatory monocytes in peripheral blood (Fig. 2A). Furthermore, M1-type macrophage infiltration in the spleen and kidney was elevated following IL-17B deletion

(Fig. 2B and C). Thus, macrophages were believed to play a critical role in injury. To further validate this, clodronate liposomes were used to deplete macrophages, followed by CFT073 infection. The depletion efficiencies are shown in Fig. S1A through D. We observed a reduction trend in creatinine level and marked alleviation of kidney and bladder injury in clodronate liposome-treated IL-17B$^{-/-}$ mice (Fig. 2D through G). Previous research has shown that macrophages regulate neutrophil infiltration in the bladder after UPEC infection (37). Therefore, we assessed neutrophil levels in peripheral blood, spleen, and kidney. Compared to infected WT mice, IL-17B$^{-/-}$ mice exhibited an increase in the percentage increased percentage of neutrophils in the kidney. But there has been little change in the neutrophil percentages in the peripheral blood and spleen (Fig. 3A through C). In addition, we analyzed the percentages of dendritic cells, NK cells, B cells, and T cells in peripheral blood, spleen, and kidney. Following CFT073 infection, various immune cells were elevated in the kidneys of IL-17B$^{-/-}$ mice compared to WT mice (Fig. S2A through F). As this study focused on acute UTIs, our attention was directed toward innate immune cells, particularly macrophages and neutrophils. To further investigate the role of neutrophils, we used anti-Ly6G neutralizing antibodies to treat mice, followed by CFT073 infection. The efficiencies of neutralizing antibodies are shown in Fig. S2G through I. We observed that the creatinine level in the plasma of mice decreased (Fig. 3D). Meanwhile, there was a reduction trend in kidney and bladder injury following neutrophil depletion in IL-17B$^{-/-}$ mice (Fig. 3E through G). These findings suggest that IL-17B deletion promotes the infiltration of various immune cells, especially macrophages, into the kidney, which is associated with exacerbated renal injury.

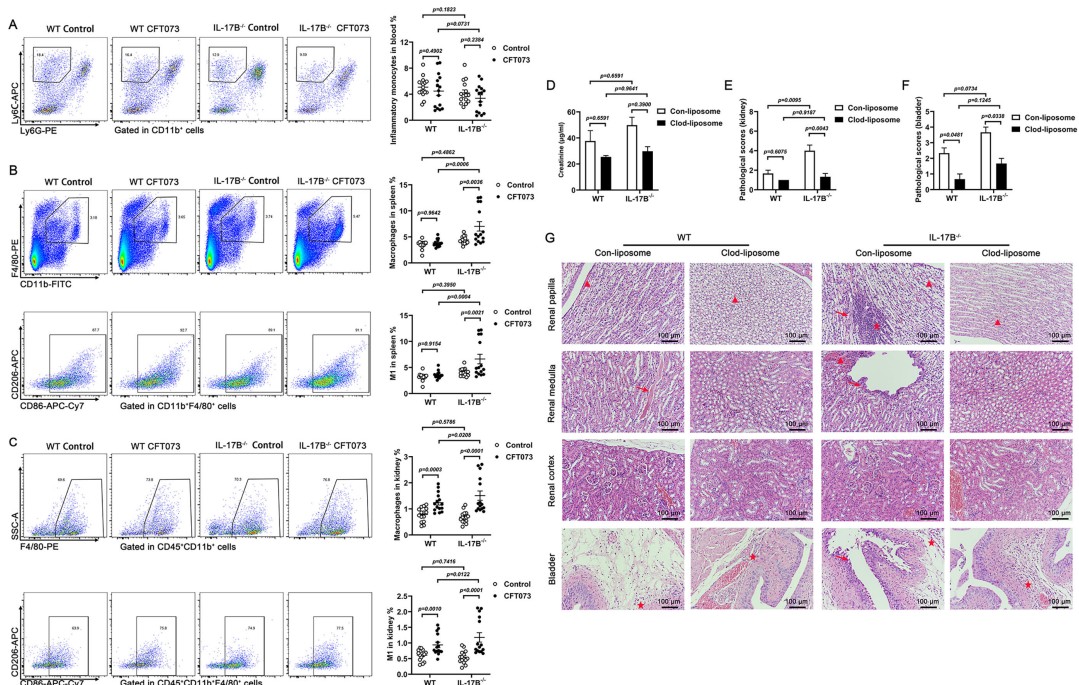

**FIG 2** Macrophages play an important role in IL-17B$^{-/-}$-induced kidney injury of mice. WT and IL-17B$^{-/-}$ female mice were infected intraurethrally with $1 \times 10^8$ CFU of CFT073 twice at 3 h intervals. (A–C) Representative flow cytometry plots and quantification of inflammatory monocytes in peripheral blood (A) and macrophages in the spleen (B) and kidney (C) after 24 h. Separate blood cells were analyzed with CD11b, followed by Ly6C and Ly6G. The M1-type macrophages in the spleen were gated on CD11b$^+$ F4/80$^+$ cells. In the kidney, the macrophages were gated on CD45$^+$ CD11b$^+$ cells, and M1-type macrophages were gated on CD45$^+$ CD11b$^+$ F4/80$^+$ cells ($n = 15$). (D–G) The mice were treated intravenously with Clod-liposomes or Con-liposomes for 24 h and then infected intraurethrally with $1 \times 10^8$ CFU CFT073 for 24 h. (D) The plasma creatinine level in mice at 24 h post-infection ($n = 3$). (E–G) H&E staining-based kidney and bladder injury scores and representative histological images. Triangles indicate hemorrhage, arrows denote tissue damage, and asterisks represent inflammatory cell infiltration. 200×, 100 µm ($n = 3$). Data are presented as mean ± SEM. Statistical comparisons (A–F) were performed using two-way ANOVA. All experiments were conducted in triplicate.

## Recombinant IL-17B alleviates renal injury in mice by influencing macrophage infiltration

Given that IL-17B deficiency worsens renal injury through enhanced macrophage infiltration, we investigated whether rIL-17B could alleviate renal damage caused by CFT073 infection. IL-17B$^{-/-}$ mice treated with rIL-17B showed significantly reduced renal injury and a reduced trend in bladder injury (Fig. 4A and B). Meanwhile, there was lower UPEC colonization in the kidneys and bladders (Fig. 4C and D). The levels of plasma creatinine were also decreased in IL-17B$^{-/-}$ mice (Fig. 4E). Flow cytometry analysis revealed that the percentages of M1 macrophages and neutrophils in the kidneys and spleens were significantly decreased with rIL-17B treatment (Fig. 4F and G). However, no significant difference was observed in inflammatory monocytes and neutrophils in the blood between NS and rIL-17B-treated groups (Fig. 4H). In the WT mice treated with rIL-17B, the same phenomenon was observed. Treatment with rIL-17B could alleviate kidney damage (Fig. S2C). The percentage of M1 macrophages in the kidneys was significantly decreased in rIL-17B-treated WT mice. And the percentage of neutrophils in the kidneys showed a decreasing trend with rIL-17B treatment (Fig. S2D). Additional analysis of immune cell subsets in the peripheral blood and kidney showed that only T cells and CD8$^+$ T cells were increased in the kidneys of rIL-17B-treated mice, consistent with changes observed in CFT073-infected IL-17B$^{-/-}$ mice. After treatment with rIL-17B, there was no significant reduction in the dendritic cells, NK cells, or neutrophils (Fig. S2J). As this study focused on acute urinary tract infections, we particularly focused on innate immune cells. These results indicate that rIL-17B mitigates UPEC-induced renal injury by regulating macrophage recruitment and suggest that IL-17B may serve as a potential therapeutic agent for urinary tract infections.

## IL-17B upregulates the expression of cytokines involved in macrophage recruitment

To explore the downstream mechanisms by which IL-17B regulates macrophage recruitment, we analyzed the expression of inflammatory cytokines and monocyte chemokines in the kidneys of UPEC-infected mice. qPCR and ELISA results showed rising

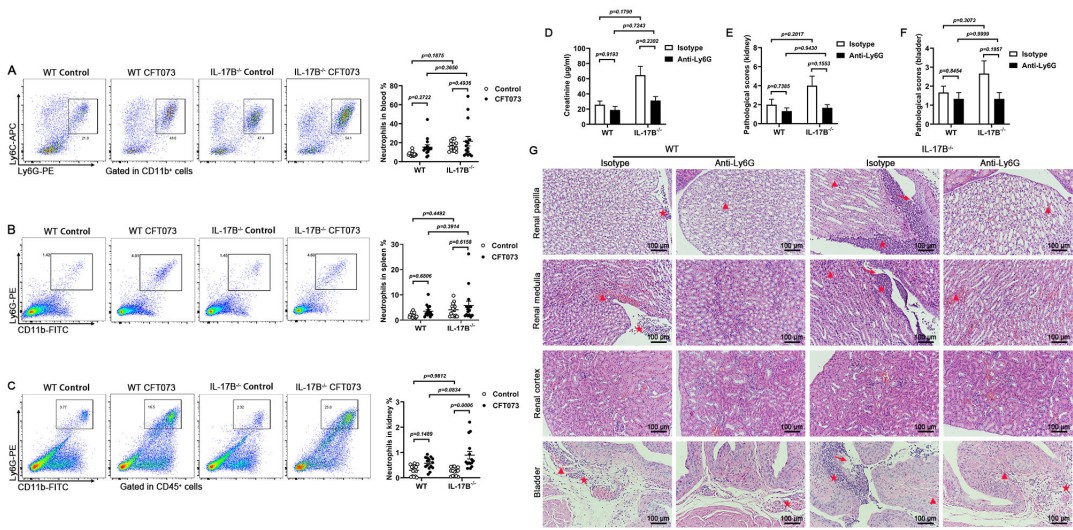

**FIG 3** Neutrophils significantly increase in the kidney of IL-17B$^{-/-}$ mice during CFT073 infection. WT and IL-17B$^{-/-}$ mice were infected with $1 \times 10^8$ CFU of CFT073, administered twice. (A–C) Proportions of neutrophils in the peripheral blood, spleen, and kidney of mice. The neutrophils in the kidney were gated on CD45$^+$ cells. (D–G) The mice were treated intravenously with isotype or anti-Ly6G antibody and infected intraurethrally with $1 \times 10^8$ CFU CFT073 for 24 h. (D) Plasma creatinine levels in mice at 24 h post-infection ($n = 3$). (E–G) The pathological scores were based on H&E staining and representative histological images of the kidney or bladder. Triangles denote hemorrhage, arrows denote tissue damage, and stars indicate inflammatory cell infiltration. 200×, 100 µm ($n = 3$). Data are presented as mean ± SEM. Statistical comparisons (A–F) were performed using two-way ANOVA. All experiments were conducted in triplicate.

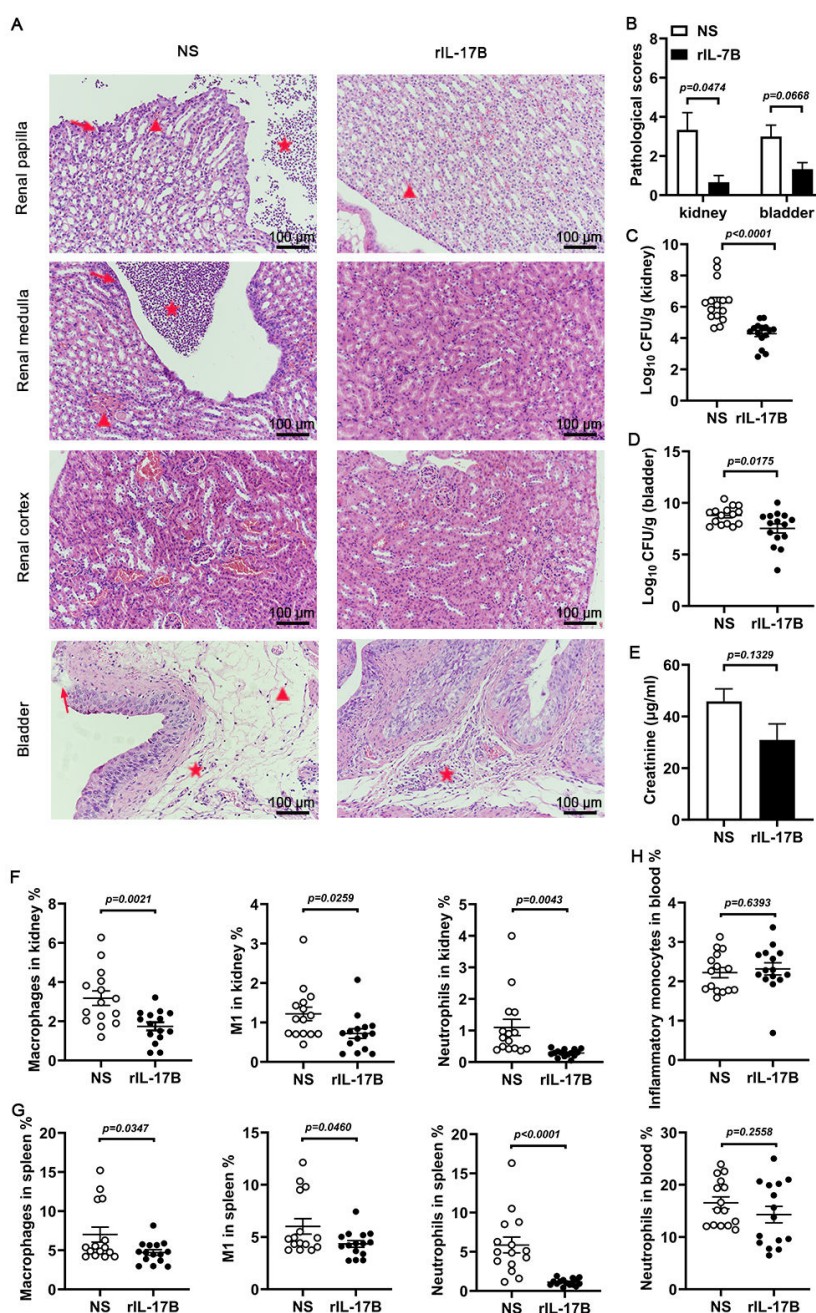

**FIG 4** IL-17B recombinant protein therapy can significantly relieve kidney damage. IL-17B$^{-/-}$ mice were treated with rIL-17B and infected with $1 \times 10^8$ CFU of CFT073, administered twice at 3 h intervals. (A and B) Representative H&E-stained kidney and bladder sections and histopathological injury scores. Triangles denote hemorrhage, arrows indicate tissue damage, and stars mark inflammatory cell infiltration. 200×, 100 μm ($n = 3$). (C and D) Bacterial colonization in the kidneys or bladder at 24 h after infection ($n = 15$). (E) The level of creatinine in mouse plasma after 24 h post-infection ($n = 3$). (F–H) Percentages of inflammatory monocytes and neutrophils in peripheral blood and macrophages and neutrophils in spleens and kidneys ($n = 15$). Data are presented as mean ± SEM. Statistical comparisons were conducted using (B) two-way ANOVA or (C–H) $t$-test. All experiments were performed in triplicate.

trends of IL-1β, IL-6, and IL-12/23 p40 in IL-17B$^{-/-}$ mice (Fig. 5A and B). After CFT073 infection, several macrophage-recruiting chemokines, including IL-1α, IL-6, IL-8 (CXCL8), G-CSF, CCL2, CCL3, CCL4, CCL7, CCL12, CCL19, CCL20, CCL22, CCL24, and CXCL14, were

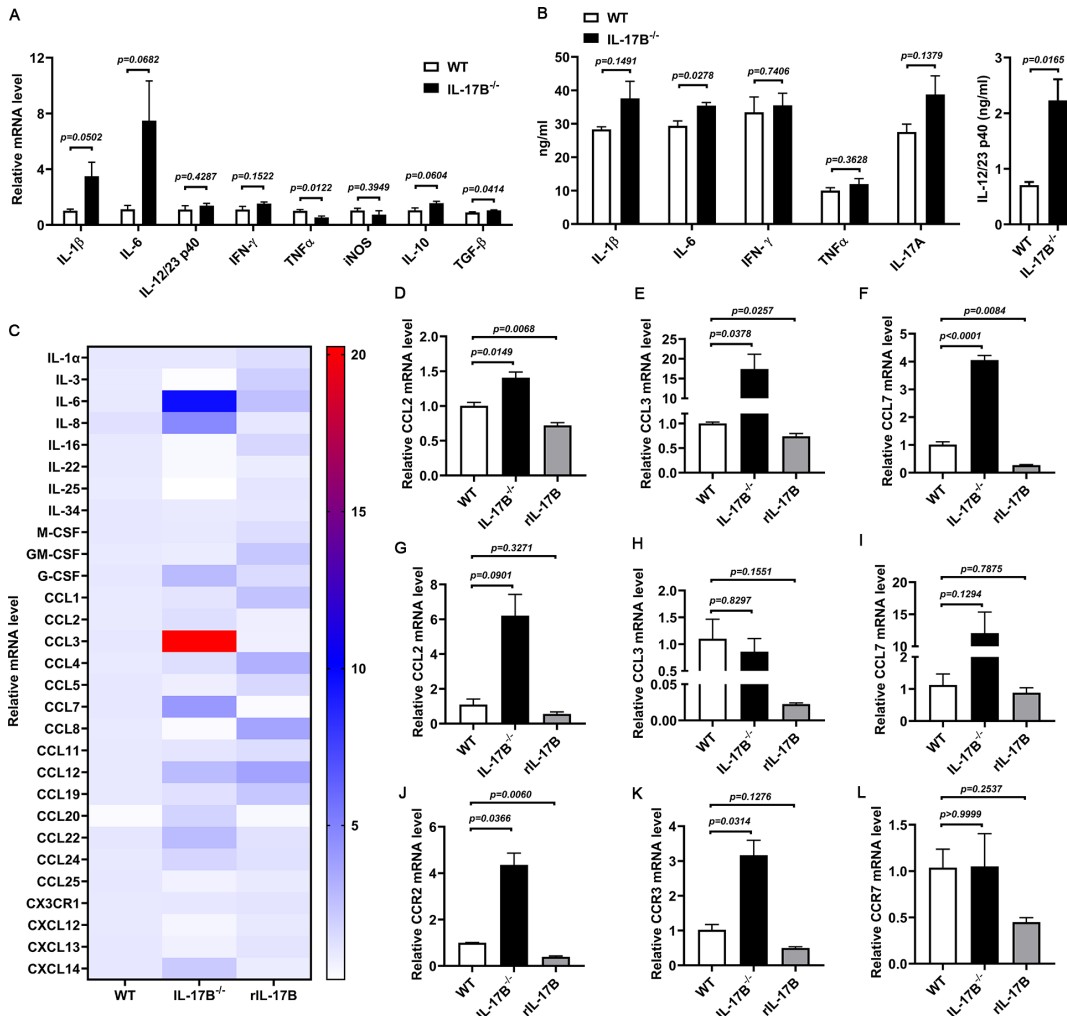

**FIG 5** IL-17B affects macrophage infiltration by influencing the expression of chemokines. WT and IL-17B$^{-/-}$ mice were infected with $1 \times 10^8$ CFU of CFT073, and a subset of WT mice received rIL-17B treatment. (A and B) qPCR and ELISA results showing pro-inflammatory cytokine expression ($n = 3$). (C–F) qPCR analysis of monocyte chemokines, including CCL2, CCL3, and CCL7 ($n = 3$). (G–I) BMDMs from WT mice and IL-17B$^{-/-}$ mice were stimulated with CFT073 [multiplicity of infection (MOI) = 0.01] for 6 h. The rIL-17B-treated group included BMDMs from WT mice stimulated by rIL-17B and CFT073. The expression levels of CCL2, CCL3, CCL7, CCR2, CCR3, and CCR7 detected by qPCR ($n = 3$). Data are shown as mean ± SEM. A and B are analyzed by $t$-test, and C–L are analyzed by one-way ANOVA. All experiments were conducted at least in triplicate.

upregulated in the kidneys of IL-17B$^{-/-}$ mice. However, treatment of CFT073-infected WT mice with rIL-17B reduced the levels of IL-22, CCL2, CCL3, and CCL7 (Fig. 5C through F; Fig. S4A). Although there was a slight decrease in CCL3 in rIL-17B-treated WT mice compared with WT mice, the expression level of CCL3 significantly increased in IL-17B$^{-/-}$ mice. We believe that CCL3 also plays a crucial role in the IL-17B regulation process. In future studies, we will further explore this. Meanwhile, we performed qPCR analysis to evaluate the expression levels of macrophages at CCL2, CCL3, CCL7, CCR2, CCR3, and CCR7 in BMDMs from WT mice, IL-17B$^{-/-}$ mice, and rIL-17B-treated mice. Compared with the BMDMs from WT mice, the levels of CCR2 and CCR3 of BMDMs from IL-17B$^{-/-}$ mice significantly increased, while the levels of CCR2 significantly decreased in BMDMs from rIL-17B-treated WT mice. At the same time, CCR3 and CCR7 in BMDMs of rIL-17B-treated WT mice have a decreased trend (Fig. 5G through I). These findings suggest that IL-17B attenuates macrophage infiltration by downregulating CCL2, CCL3, and CCL7 expression, thereby mitigating renal injury. In conclusion, IL-17B reduces kidney damage induced by

bacterial infection, though inhibiting the production of key chemokines (CCL2, CCL3, and CCL7), highlighting its potential as a therapeutic target for renal injury (Fig. 6).

## DISCUSSION

Nosocomial and community-acquired infections are the two primary routes of UTIs. Infection with UPEC not only causes recurrent episodes but also results in structural damage to the urinary system, including the bladders and kidneys. Due to the shorter female urethra, UTIs are common among women. Approximately 50% of women will experience at least one UTI during their lifetime. Among those women with a first infection, 5%–25% will develop recurrent UTIs. The recurrence rate increases significantly with growing age, especially in postmenopausal women. Patients with recurrent or severe UTIs not only face physical and economic burdens but may also lead to tissue damage and sequelae resulting from pyelonephritis (7, 8, 38). Therefore, there is an urgent need to develop novel therapeutic strategies to mitigate kidney damage.

The host immune response is often accompanied by immune cell infiltration and cytokine secretion. Previous studies have shown that excessive inflammation can lead to tissue injury, although immune cells also participate in tissue repair following immune damage (39). As an important member of the IL-17 family, the role of IL-17B in inflammation responses, particularly in bacterial infections, remains poorly defined. *In vitro* studies using porcine jejunal epithelial cells transfected with plasmids encoding IL-17B and IL-25, followed by bacterial infection with *Pseudomonas aeruginosa*, *Streptococcus pyogenes*, enterohemorrhagic *Escherichia coli*, and *Shigella sonnei*, demonstrated significant upregulation of IL-17B and IL-25. Elevated IL-17B levels promoted antimicrobial peptide production and inhibited microbial invasion (40). In *Helicobacter pylori* infection, IL-17RB expression in the gastric mucosa is markedly reduced in both patients and mice. The *Helicobacter pylori* virulence factor CagA downregulates IL-17RB via the PI3K/AKT signaling pathway in gastric epithelial cells. However, IL-17RB facilitates CD11b$^+$CD11c$^-$ myeloid cell accumulation through binding to IL-25, thereby suppressing

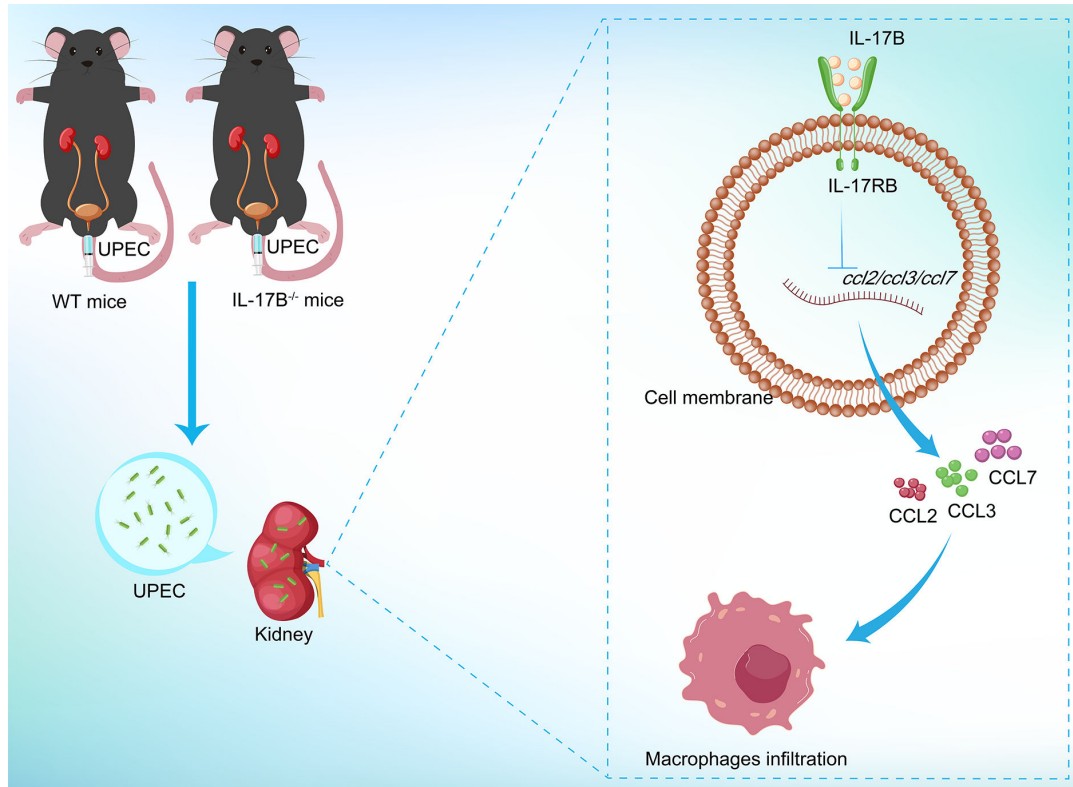

**FIG 6** Mechanism by which IL-17B influences macrophage infiltration to alleviate kidney damage.

bacterial colonization in the gastric mucosa (41). In a model of acute colitis induced by *Citrobacter rodentium*, IL-17B deficiency exacerbates inflammation, causes a significant decrease in mouse body weight, and increases bacterial colonization (21). These findings support a protective role of IL-17B in bacterial infection-induced inflammation, consistent with the present study. In contrast, in a bleomycin-induced pulmonary fibrosis model, bacteria from the *Bacteroides* and *Prevotella genera* promote pulmonary fibrosis by inducing IL-17B, which recruits immune cells and triggers the expression of fibrosis-related proteins, indicating a pro-inflammatory function (42). These conflicting results suggest that the role of IL-17B in inflammation may vary depending on the disease context. The present study aimed to clarify the function of IL-17B in UTIs, particularly its role in kidney injury.

Our results show that IL-17B and IL-17RB expression levels significantly increase in the kidneys during UTIs. The mortality rate of IL-17B$^{-/-}$ mice increased significantly, and kidney damage was more severe, indicating that IL-17B might be involved in the pathogenesis of kidney injury during UTIs. Previous research has highlighted the involvement of multiple immune cell types in the host response to UTIs (43, 44), with macrophages playing a crucial role in the course of UPEC infection. Macrophages are essential for bacterial clearance, yet excessive macrophage activity can lead to tissue injury. Notably, depletion of macrophages alleviates kidney injury during UTIs (36, 45–47). After depleting phagocytes, the degree of kidney injury in mice was significantly reduced. However, when infected IL-17B$^{-/-}$ mice were treated with clodronate liposomes, the proportion of monocytes in the peripheral blood increased instead. This phenomenon was inconsistent with the situation in WT mice. Therefore, we speculate that IL-7B may affect the phagocytosis of monocytes rather than macrophages. The specific mechanism will be further explored in our future research.

In this study, we found that IL-1α, IL-6, IL-8, G-CSF, CCL2, CCL3, CCL4, CCL7, CCL12, CCL19, CCL20, CCL22, CCL24, and CXCL14 levels increased in the kidneys of IL-17B$^{-/-}$ mice following infection. Treatment with recombinant IL-17B (rIL-17B) reduced the expression of IL-22, CCL2, CCL3, and CCL7, suggesting that IL-17B modulates macrophage infiltration by downregulating these chemokines. Meanwhile, IL-17B deficiency led to elevated IL-1β and IL-6 levels, along with increased bacterial colonization in the kidneys. These findings indicate that IL-17B plays an anti-inflammatory role in UPEC-induced UTIs. We also observed that changes in neutrophil levels mirrored those of macrophages. The absence of IL-17B resulted in a significant increase in neutrophils, consistent with previous findings that macrophages regulate neutrophil migration to infected bladder tissues (37). This suggests that IL-17B may reduce neutrophil infiltration indirectly by suppressing macrophage infiltration in infected kidneys. However, studies on sciatic nerve injury have shown that IL-17B activates IL-17RB and induces the release of chemokines, including CCL2, CCL3, CCL19, CCL22, and G-CSF, facilitating macrophage recruitment and promoting nerve regeneration (24). In pulmonary fibrosis, IL-17B secreted by macrophages has been shown to promote neutrophil migration and act as a pro-inflammatory mediator, which promotes the formation of pulmonary fibrosis (24, 42). This may be related to the repair of nerve damage and the formation of pulmonary fibrosis, both of which are closely associated with M2-type macrophages. However, in acute infections caused by UPEC, macrophages are often associated with M1-type macrophages. Therefore, we hypothesize that IL-17B exerts differential effects on distinct macrophage subtypes (48, 49). The contrasting roles of IL-17B in neutrophil regulation across different diseases warrant further investigation, particularly regarding the contributions of chemokines or IL-25. These findings all indicate that IL-17B has different or even opposite effects in different inflammatory responses, which may be related to the distinct pathogenesis of different diseases. For example, IL-17B can also inhibit IL-25-mediated IL-6 production in colitis (20), which is consistent with our finding in this study that the level of IL-6 in the kidneys of mice was also increased after the absence of IL-17B.

Additionally, IL-17B has been implicated in tumorigenesis. In pancreatic, gastric, breast, and lung cancers, IL-17B is closely related to the proliferation and migration of cancer cells. Its overexpression is strongly associated with postoperative metastasis and poor prognosis (50–54). In this study, IL-17B expression was significantly upregulated during the acute phase of UPEC infection. Whether IL-17B plays a similar role in chronic inflammation remains unknown. Since chronic inflammation can lead to tumorigenesis, the potential involvement of IL-17B in inflammation-associated cancer progression requires further study.

Based on our research results, we investigated whether treating infected WT mice or IL-17B-deficient mice can alleviate renal injury in mice. Those results suggest that IL-17B can be a potential factor for treating renal injury caused by urinary tract infection. However, cytokine therapy is often subject to many restrictions due to the short half-life of cytokines. Meanwhile, the kidneys have a filtering function, causing poor efficacy when using cytokine therapy on the kidneys. Pegylated cytokines, a novel method, can effectively reduce cytokine clearance caused by renal metabolism. But the production of anti-PEG antibodies can bring about a series of toxic side effects. Meanwhile, cytokines have high efficiency. Therefore, large-dose delivery may trigger a strong immune response or suppression in the body. Thus, we think that using nanoparticles or lipid particles with small doses and multiple deliveries may be safer. At the same time, delivering IL-17B mRNA can also be considered to extend the half-life of the drug (55–58). These aspects warrant further investigation in future studies.

## ACKNOWLEDGMENTS

We gratefully acknowledge Professor Quan Wang (Institute of Medicinal Biotechnology, Chinese Academy of Medical Sciences & Peking Union Medical College) and Qingli Bie (Affiliated Hospital of Jining Medical University) for generously providing the bacterial strain CFT073 and IL-17B KO mice used in this study.

This work was supported by the National Natural Science Foundation of China (Nos 82101864 and 82171810), Shandong Province College Youth Innovation Team Development Plan (No. 2023KJ262), and the College Students' Innovative Entrepreneurial Training Project of Jining Medical University (cx2024219z).

Conceptualization, H.X.; data curation, H.X. and C.W.; methodology, C.W., M.L., L.W., and Y.Z.; software and formal analysis, C.W. and Z.N.; writing—original draft preparation, C.W.; writing—review and editing, H.X. and X.L.; investigation, H.X. and X.L.; funding acquisition, H.X. and C.W. All authors have read and agreed to the published version of the manuscript.

## AUTHOR AFFILIATIONS

[1]Institute of Immunology and Molecular Medicine, Jining Medical University, Jining, China
[2]Jining Key Laboratory of Immunology, Jining Medical University, Jining, China

## AUTHOR ORCIDs

Changying Wang  http://orcid.org/0000-0002-3024-6664
Zihan Niu  http://orcid.org/0009-0000-9173-8059
Huabao Xiong  http://orcid.org/0000-0003-0180-2104

## FUNDING

| Funder | Grant(s) | Author(s) |
| --- | --- | --- |
| National Natural Science Foundation of China | 82101864 | Changying Wang |
| National Natural Science Foundation of China | 82171810 | Huabao Xiong |

| Funder | Grant(s) | Author(s) |
|---|---|---|
| Shandong Provincial Youth Innovation Technology Support Program | 2023KJ262 | Changying Wang |

## AUTHOR CONTRIBUTIONS

Changying Wang, Conceptualization, Data curation, Formal analysis, Funding acquisition, Methodology, Project administration, Software, Writing – original draft, Writing – review and editing | Min Liu, Methodology | Luyue Wang, Methodology | Yixin Zhang, Methodology | Zihan Niu, Writing – original draft | Xue Liu, Investigation, Writing – review and editing | Huabao Xiong, Conceptualization, Data curation, Funding acquisition, Investigation, Writing – review and editing

## DATA AVAILABILITY

The original contributions proposed in this paper are fully documented within the manuscript. Any further inquiries can be directed to the corresponding author.

## ETHICS APPROVAL

This study was conducted with institutional regulations for animal care and approved by the Animal Ethics Committee of Jining Medical University (Ethics Review Number: JNMC-2023-DW-133).

## ADDITIONAL FILES

The following material is available online.

### Supplemental Material

**Supplemental materials (Spectrum02244-25-s0001.pdf).** Fig. S1 to S4, Tables S1 and S2, and the list of primers and reagents.

### Open Peer Review

**PEER REVIEW HISTORY (review-history.pdf).** An accounting of the reviewer comments and feedback.

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
