## [Reviewer comments · Microbiology Spectrum]

Microbiology Spectrum

IL-17B protects against uropathogenic *E. coli*-induced kidney injury via macrophage infiltration modulation

Changying Wang, Min Liu, Luyue wang, Yixin Zhang, Zihan Niu, Xue Liu, and Huabao Xiong

Corresponding Author(s): Huabao Xiong, Jining Medical University

Review Timeline:

Submission Date:	July 22, 2025
Editorial Decision:	August 18, 2025
Revision Received:	November 4, 2025
Editorial Decision:	December 4, 2025
Revision Received:	January 18, 2026
Editorial Decision:	February 5, 2026
Revision Received:	February 27, 2026
Accepted:	March 20, 2026

Editor: Gregory Wiedman

Reviewer(s): Disclosure of reviewer identity is with reference to reviewer comments included in decision letter(s). The following individuals involved in review of your submission have agreed to reveal their identity: Basel H Abuaita (Reviewer #1); Parmanand Prabhakar (Reviewer #2)

Transaction Report:

DOI: <https://doi.org/10.1128/spectrum.02244-25>

Re: Spectrum02244-25 (IL-17B protects uropathogenic *E. coli* induced kidney injury by modulating macrophage infiltration)

Dear Prof. Huabao Xiong:

Thank you for the privilege of reviewing your work. Below you will find my comments, instructions from the Spectrum editorial office, and the reviewer comments.

Revision Guidelines

Sincerely,
Gregory Wiedman
Editor
Microbiology Spectrum

Reviewer #2 (Comments for the Author):

The manuscript investigates the role of IL-17B in protecting against uropathogenic *Escherichia coli* (UPEC)-induced kidney injury through modulation of macrophage infiltration. The study addresses a novel aspect of host immune response in urinary tract infection (UTI) pathogenesis, providing both mechanistic insight and potential translational relevance. The experiments are well-planned, and the results are supported by multiple complementary approaches. However, several points require

clarification, additional discussion, and language refinement to improve scientific rigor and readability.

The manuscript by Wang et al investigated the role of IL-17B signaling in immunity against urinary tract infections (UTIs) caused by Uropathogenic *Escherichia coli* (UPEC). The main result suggests that IL-17B knockout (KO) mice are more susceptible to UPEC infection compared to wild-type (WT) mice. The data showed that IL-17B KO mice survived less, had higher bacterial burdens in the kidneys, and exhibited more severe damaged kidneys when compared to WT mice. Furthermore, immunophenotype analysis suggests that inflammatory M1 and neutrophils accumulated in the kidneys of infected IL-17B KO mice more than WT mice. The authors conclude that the increased susceptibility of IL-17B KO mice could be due to the increased accumulation of these inflammatory cells in the kidneys. Defining the contribution of IL-17B signaling to immunity against UPEC will incrementally increase our understanding of host defense against UPEC. However, several issues diminished the overall enthusiasm and confidence that the data would advance the field. First, the route of bacterial inoculation is unknown, which is essential for the interpretation of the results. The abstract indicates the bacteria are inoculated intranasally (line 18), while the Figure 1 legend states the bacteria are inoculated via intraurethral. Yet, the materials and methods section did not include any information on how the mice were infected. Second, the inoculation size seems to be too high compared to other studies that used the magnitude of 2 log less bacteria (PMID: 20083670). Third, the n number for each experiment is not clearly stated. Stating that experiments were performed in triplicate is insufficient to conclude from *in vivo* experiments. Usually, the *in vivo* data, like CFUs, immunophenotyping, and survival, must be performed using n=15 mice from each group over three experiments that are performed on different days. Lastly, most of the representative flow cytometry histograms don't match the corresponding quantified graphs, which often contradict the authors' conclusions. Because of all these issues, I found that the manuscript is not suitable for publication in its current state. I also provided additional concerns that must be addressed.

Major Concerns:

1. It is not clear what the rationale is for studying the IL-17B signaling. It is already shown that IL-17A signaling is essential for immunity against UPEC. What about IL-17C signaling? To formulate a better rationale, the authors must quantify the expression of most of the IL-17 family effectors, including IL-17A, IL-17B, IL-17C, IL-17D, IL-17E, IL-17RA, IL-17RC, IL-17RB, IL-17RE, and IL-17RD.
2. What do the DAP12^{-/-} mice in the legend of Figure 2 represent? This seems to be part of another study.
3. The pathological score of the histology slides is not clearly stated. Scoring using damage is insufficient. What are the criteria the authors used to rank the damaged tissue from 1 to 4? Is it based on cell death, immune cell infiltrations, or tubular cell sloughing? The authors can also quantify the serum creatinine levels.
4. Statistical analysis for Figure 1D, Figure 2 panels, and Figure 3 panels must be performed using Two-way ANOVA and not the Student t-test.

5. Flow cytometry histograms don't match the quantified graphs in Figures 2 and 3. For example, Figure 2A showed that 17.1 percent of blood cells are monocytes from WT control mice, but the graph showed approximately 3%. Also, the flow cytometry histograms in Figure 2B indicated that the percentage of macrophages in the spleen of uninfected IL-17B KO mice was more than that of wild type. Thus, the increase in macrophages is independent of the infection. The authors must correct their conclusion and provide details on how they performed the calculations. It is also essential to quantify the absolute counts rather than relative percentages. Lastly, the authors must check whether IL-17B KO mice have some sort of inflammation. The authors should check cytokine levels in the serum of uninfected mice.
6. Again, based on flow cytometry data, there are more relative inflammatory macrophages in the kidney of infected wild-type mice (92% of 66%= 60%) compared to the kidney of IL-17B KO (85% of 45% = 38%) mice. This does not match the graph and contradicts the author's conclusion.
7. The authors must define the contribution of neutrophils and monocytes to the susceptibility of IL-17B KO mice. This can be done by depleting the specific cell population in IL-17B KO mice before infection. This is essential to conclude whether the increase in inflammatory neutrophils, monocytes, and/or monocyte-derived macrophages was required for the severity of infection in the IL-17B KO mice.
8. In Figure 4, WT mice were treated with recombinant protein rIL-17B, but IL-17B KO mice were not. Since WT mice already produce the target cytokine endogenously, it is unclear how supplementing rIL-17B exogenously will be beneficial. Also, the route of injecting rIL-17B is not stated, which is essential to evaluate the data. This experiment should be performed in IL-17B KO mice to assess whether rIL-17B treatment will protect these mice from UPEC-induced kidney injury.
9. For a UTI study, bladder CFUs and histology are expected. The authors must provide these data.

Minor Concerns:

1. The sentence in lines 211-212 must be referenced.
2. References 4 and 5 don't support the information stated in lines 60-63. The authors must provide different references.
3. In Figure 1B, the authors stated that the experiment was performed for 24 hours, and the x-axis only shows 20 hours. What happens to infected wild-type mice beyond the 20 hours? Do they eventually die due to infection? The n number for each mouse group must be indicated in the figure.

4. The authors concluded in lines 213-215 that the IL-17B KO mice had a higher percentage of inflammatory monocytes but failed to report the statistical significance in the Figure 2A.

Review Remarks

- ✓ The title is accurate but could be made more concise and impactful by highlighting the therapeutic implication, e.g., "*IL-17B Protects against Uropathogenic E. coli-Induced Kidney Injury via Macrophage Infiltration Modulation*".
- ✓ Abstract is informative but **too dense**; consider shortening and focusing on the most critical findings for better readability.
- ✓ The infection route is stated as “intranasally infected” (line 17) — this seems unusual for a UTI model and may require clarification or correction.
- ✓ Dose rationalization: While 10^8 and 10^9 CFU doses are mentioned, a brief justification for dose selection would strengthen reproducibility.
- ✓ The flow cytometry panel description is long; moving antibody details to supplementary material would improve readability.
- ✓ Some p-values are given as “ $P = 0.067$ ” without interpretation—should clarify whether considered biologically relevant despite not meeting conventional significance.
- ✓ Figures would benefit from exact p-values or confidence intervals rather than just asterisks.
- ✓ While the discussion addresses context-specific roles of IL-17B, it could further explore **therapeutic translation challenges**, such as delivery methods, dosing, and possible off-target effects.
- ✓ The sex-specific immune responses to UPEC (mentioned in background) are not revisited in the results or discussion—this is an opportunity to enhance the narrative.
- ✓ More emphasis could be placed on how findings fit with other IL-17 family cytokine research.
- ✓ Minor grammatical inconsistencies and typographical errors (e.g., “institUTISonal” in line 361) should be corrected.
- ✓ Some sentences are overly long; breaking them up would improve readability.

Recommendation: Minor revision before acceptance mainly for clarity, methodological transparency, and narrative tightening.

Dear Editor and Reviewers:

We would like to express our sincere gratitude to you for your continued attention to our manuscript titled " IL-17B protects against uropathogenic *E. coli*-induced kidney injury via macrophage infiltration modulation " (submission number: Spectrum02244-25R1). We are deeply grateful to the editors and reviewers for your meticulous revisions and corrections of our work. During this revision process, we carefully revised our manuscript based on the valuable suggestions provided by the reviewers, which helped us better control the details of our manuscript. The revised manuscript has been uploaded. We sincerely appreciate any comments and suggestions from the reviewers for further improvement of our manuscript. Based on your constructive comments, we have made corrections and modifications to the revised manuscript. The detailed revisions are outlined below.

Reviewer #1:

The manuscript by Wang et al investigated the role of IL-17B signaling in immunity against urinary tract infections (UTIs) caused by Uropathogenic *Escherichia coli* (UPEC). The main result suggests that IL-17B knockout (KO) mice are more susceptible to UPEC infection compared to wild-type (WT) mice. The data showed that IL-17B KO mice survived less, had higher bacterial burdens in the kidneys, and exhibited more severe damaged kidneys when compared to WT mice. Furthermore, immunophenotype analysis suggests that inflammatory M1 and neutrophils accumulated in the kidneys of infected IL-17B KO mice more than WT mice. The authors conclude that the increased susceptibility of IL-17B KO mice could be due to the increased accumulation of these inflammatory cells in the kidneys. Defining the contribution of IL-17B signaling to immunity against UPEC will incrementally increase our understanding of host defense against UPEC. However, several issues diminished the overall enthusiasm and confidence that the data would advance the field. First, the route of bacterial inoculation is unknown, which is essential for the interpretation of the results. The abstract indicates the bacteria are inoculated intranasally (line 18), while the Figure 1 legend states the bacteria are inoculated via intraurethral. Yet, the materials and methods section did not include any information on how the mice were infected. Second, the inoculation size seems to be too high compared to other studies that used the magnitude of 2 log less bacteria (PMID: 20083670). Third, the n number for each experiment is not clearly stated. Stating that experiments were performed in triplicate is insufficient to conclude from *in vivo* experiments. Usually, the *in vivo* data, like CFUs, immunophenotyping, and survival, must be performed using n=15 mice from each group over three experiments that are performed on different days. Lastly, most of the representative flow cytometry histograms don't match the corresponding quantified graphs, which often contradict the authors' conclusions. Because of all these issues, I found that the manuscript is not suitable for publication in its current state. I also provided additional concerns that must be addressed.

Response to comment: Dear Reviewer, we sincerely appreciate your valuable and

professional suggestions. The corrections for this section are as follows:

1. The mice were infected transurethrally. We apologize for the mistake we have made.

We have corrected it in the abstract, and the transurethral infection was indicated in the material method (line 117)

2. When CFT073 is infected with the C57BL/6 background mice, the amount of infection required ranges from 1×10^7 to 1×10^9 CFU/mouse. And when we infect at high concentrations (1×10^9 CFU/mouse), the mortality rate of mice is higher. So, we chose the concentration of 1×10^8 CFU/mouse for infection. Meanwhile, 1×10^8 CFU/mouse is more widely used in the study of pyelonephritis (1-3). We have marked these cited references at 106.

(1) Li K, Wu K-Y, Wu W, Wang N, Zhang T, Choudhry N, Song Y, Farrar CA, Ma L, Wei L-l. 2017. C5aR1 promotes acute pyelonephritis induced by uropathogenic *E. coli*. *JCI insight* 2: e97626.

(2) Yang H, Li Q, Wang C, Wang J, Lv J, Wang L, Zhang Z-S, Yao Z, Wang Q. 2018. Cytotoxic necrotizing factor 1 downregulates CD36 transcription in macrophages to induce inflammation during acute urinary tract infections. *Frontiers in Immunology* 9:1987.

(3) de Dios Ruiz-Rosado J, Robledo-Avila F, Cortado H, Rangel-Moreno J, Justice SS, Yang C, Spencer JD, Becknell B, Partida-Sanchez S. 2021. Neutrophil-macrophage imbalance drives the development of renal scarring during experimental pyelonephritis. *Journal of the American Society of Nephrology* 32:69-85.

3. We have repeated these experiments and increased the number of mice to 15 in colony counting, flow cytometry, and survival-related experiments. These modifications are presented in Fig.1-4.

4. We have revised and annotated the non-corresponding elements in Fig.2 and Fig.3, with all modifications clearly presented therein.

Major Concerns:

1.It is not clear what the rationale is for studying the IL-17B signaling. It is already shown that IL-17A signaling is essential for immunity against UPEC. What about IL-17C signaling? To formulate a better rationale, the authors must quantify the expression of most of the IL-17 family effectors, including IL-17A, IL-17B, IL-17C, IL-17D, IL-17E, IL-17RA, IL-17RC, IL-17RB, IL-17RE, and IL-17RD.

Response to comment: Thank you for the valuable suggestions from the experts. Since IL-17B is a relatively new member of the IL-17 family and its role in inflammatory responses is often dual, we aim to explore whether it can be used as a regulatory factor to control kidney damage caused by urinary tract infections. We have conducted tests and the results are shown in Fig 1. In the future, we will also continue to investigate the roles of other family members, such as IL-17D.

2.What do the DAP12^{-/-} mice in the legend of Figure 2 represent? This seems to be

part of another study.

Response to comment: We are very grateful for your meticulous review, and deeply sorry for the annotation errors we made. The modified content is shown in Fig.2 legend.

3.The pathological score of the histology slides is not clearly stated. Scoring using damage is insufficient. What are the criteria the authors used to rank the damaged tissue from 1 to 4? Is it based on cell death, immune cell infiltrations, or tubular cell sloughing? The authors can also quantify the serum creatinine levels.

Response to comment: Thank you for your careful review and professional comments. We have taken the scores according to whether there was extensive inflammatory cell infiltration, causing changes in the complete tissue morphology, and tubular atrophy. Those are shown in line 139. Meanwhile, we detected the creatinine levels in the plasma of each group. The results are shown in Fig.1F, Fig.2D, Fig.3D, and Fig.4E.

4.Statistical analysis for Figure 1D, Figure 2 panels, and Figure 3 panels must be performed using Two-way ANOVA and not the Student t-test.

Response to comment: Dear reviewer, thank you for the professional comments. We have reanalyzed all the involved groups and modified the corresponding annotations. Those revisions are shown in Fig.1D, Fig. 2, Fig. 3, and Fig.1S

5.Flow cytometry histograms don't match the quantified graphs in Figures 2 and 3. For example, Figure 2A showed that 17.1 percent of blood cells are monocytes from WT control mice, but the graph showed approximately 3%. Also, the flow cytometry histograms in Figure 2B indicated that the percentage of macrophages in the spleen of uninfected IL-17B KO mice was more than that of wild type. Thus, the increase in macrophages is independent of the infection. The authors must correct their conclusion and provide details on how they performed the calculations. It is also essential to quantify the absolute counts rather than relative percentages. Lastly, the authors must check whether IL-17B KO mice have some sort of inflammation. The authors should check cytokine levels in the serum of uninfected mice.

Response to comment: Thank you for your thorough review and constructive suggestions. We have made the necessary modifications and annotations for the cases that do not correspond. Before the experiment, we also checked the condition of the mice. We re-examined and re-analyzed the data and found that the indication of inflammation in IL-17B KO mice might be due to improper selection of images. We have corrected this issue. The details of the modifications are shown in Fig.2 and Fig.3.

We truly appreciate your valuable suggestions. The absolute count in flow cytometry does indeed have very important significance. We apologize that, due to the limitations of our flow cytometry instrument, absolute cell counts cannot be reliably obtained. Meanwhile, we attempted to estimate absolute cell numbers by measuring the concentration of single-cell suspensions using a cell counter and applying appropriate conversions. However, this method may have the problem of inaccurate

data. Here we present the numbers of macrophages and M1-type macrophages in the kidneys of WT and IL-17B KO mice after CFT073 infection, as calculated by us. Furthermore, as a percentage-based representation of flow cytometry data is widely adopted in the literature (4-5), we want to continue to display the percentage in this study. In future work, we will prioritize the use of methods enabling absolute cell quantification to enhance data robustness.

(4) Chowdhury S, Castro S, Coker C, Hinchliffe TE, Arpaia N, Danino T. 2019. Programmable bacteria induce durable tumor regression and systemic antitumor immunity. *Nat Med*. 25:1057-1063.

(5) Mulvey MA, Wang Z, Jiang Z, Zhang Y, Wang C, Liu Z, Jia Z, Bhushan S, Yang J, Zhang Z. 2024. Exosomes derived from bladder epithelial cells infected with uropathogenic *Escherichia coli* increase the severity of urinary tract infections (UTIs) by impairing macrophage function. *PLOS Pathogens* 20.

6. Again, based on flow cytometry data, there are more relative inflammatory macrophages in the kidney of infected wild-type mice (92% of 66% = 60%) compared to the kidney of IL-17B KO (85% of 45% = 38%) mice. This does not match the graph and contradicts the author's conclusion.

Response to comment: We are very grateful for the detailed review. We have adjusted the inconsistent pictures. Thank you for your careful correction. For details of the modification, please see Fig.2.

7. The authors must define the contribution of neutrophils and monocytes to the susceptibility of IL-17B KO mice. This can be done by depleting the specific cell population in IL-17B KO mice before infection. This is essential to conclude whether the increase in inflammatory neutrophils, monocytes, and/or monocyte-derived macrophages was required for the severity of infection in the IL-17B KO mice.

Response to comment: Thank you for the valuable suggestions from the experts. To classify the function of monocytes and neutrophils, we revalidated by eliminating macrophages with clodronate liposomes and depleting neutrophils with Ly6G neutralizing antibody. The results are shown in Fig.2 and Fig.3.

8. In Figure 4, WT mice were treated with recombinant protein rIL-17B, but IL-17B KO mice were not. Since WT mice already produce the target cytokine endogenously,

it is unclear how supplementing rIL-17B exogenously will be beneficial. Also, the route of injecting rIL-17B is not stated, which is essential to evaluate the data. This experiment should be performed in IL-17B KO mice to assess whether rIL-17B treatment will protect these mice from UPEC-induced kidney injury.

Response to comment: We are grateful for the valuable suggestions. We have re-conducted the experiment using rIL-17B to treat IL-17B KO mice. Thus, we moved the content of the original Fig.4 to Fig.S2.

9.For a UTI study, bladder CFUs and histology are expected. The authors must provide these data.

Response to comment: Thank you for your thorough review and constructive suggestions. We have included the content related to the bladder in this revision.

Minor Concerns:

1.The sentence in lines 211-212 must be referenced.

Response to comment: We appreciate your valuable suggestions. We have already made modifications to this part of the content. For details, please refer to line 233 of the main text.

2.References 4 and 5 don't support the information stated in lines 60-63. The authors must provide different references.

Response to comment: Thank you for your thorough review and constructive suggestions. We have replaced the references in line 55.

3.In Figure 1B, the authors stated that the experiment was performed for 24 hours, and the x-axis only shows 20 hours. What happens to infected wild-type mice beyond the 20 hours? Do they eventually die due to infection? The n number for each mouse group must be indicated in the figure.

Response to comment: We appreciate your valuable suggestions. The subsequent mice will survive. In fact, generally, no more mice will die after 48 hours. We have already marked the n, which are shown in the figure legends.

4.The authors concluded in lines 213-215 that the IL-17B KO mice had a higher percentage of inflammatory monocytes but failed to report the statistical significance in the Figure 2A.

Response to comment: Thank you for your valuable professional opinion. We have already carried out the relevant part in accordance with its description, which is shown in line 235.

Reviewer #2:

Review Remarks

The title is accurate but could be made more concise and impactful by highlighting the therapeutic implication, e.g., "*IL-17B Protects against Uropathogenic E. coli-Induced Kidney Injury via Macrophage Infiltration Modulation*".

Response to comment: Dear Reviewer, we sincerely appreciate your careful guidance and professional suggestions. This has greatly helped us improve the quality of our manuscripts. We changed the title from "*IL-17B protects uropathogenic E. coli induced kidney injury by modulating macrophage infiltration*" to "*IL-17B Protects against Uropathogenic E. coli-Induced Kidney Injury via Macrophage Infiltration Modulation*".

Abstract is informative but **too dense**; consider shortening and focusing on the most critical findings for better readability.

Response to comment: Thank you for your thorough review and constructive suggestions. We have made the revisions to the abstract. The revised content is as the abstract in the revised manuscript.

The infection route is stated as "intranasally infected" (line 17) — this seems unusual for a UTI model and may require clarification or correction.

Response to comment: We are very grateful for the detailed review. We apologize for the mistake we have made. We have removed this part from the abstract, but have explained it in the material methods.

Dose rationalization: While 10^8 and 10^9 CFU doses are mentioned, a brief justification for dose selection would strengthen reproducibility.

Response to comment: Thank you for the professional comments from the reviewers. We have added references for the selection of these doses and provided a brief explanation in line 118.

The flow cytometry panel description is long; moving antibody details to supplementary material would improve readability.

Response to comment: We are very grateful for your detailed comments and suggestions. We have moved this part to the supplementary table2.

Some p-values are given as "P = 0.067" without interpretation—should clarify whether considered biologically relevant despite not meeting conventional significance.

Response to comment: Thanks for your great suggestion on improving the accessibility of our manuscript. We have added an explanation to this part of the content. The details are shown in line 296.

Figures would benefit from exact p-values or confidence intervals rather than just

asterisks.

Response to comment: We are very grateful to the expert for the valuable suggestions.

While re-analyzing the data, we also added the p-values.

While the discussion addresses context-specific roles of IL-17B, it could further explore **therapeutic translation challenges**, such as delivery methods, dosing, and possible off-target effects.

Response to comment: Thank you for your thorough review and constructive suggestions. In our discussion, we added this part, which was shown in line 389.

The sex-specific immune responses to UPEC (mentioned in background) are not revisited in the results or discussion—this is an opportunity to enhance the narrative.

Response to comment: Thank you for the valuable suggestions from the experts.

We have added a discussion of this aspect in the revised manuscript in line 307.

More emphasis could be placed on how findings fit with other IL-17 family cytokine research.

Response to comment: We are grateful for the valuable suggestions. Due to the dual nature of IL-17B, and given that we focused on its anti-inflammatory effect. We only explored its correlation with IL-25, which shares the same receptor as IL-17B.

Minor grammatical inconsistencies and typographical errors (e.g., “institUTISonal” in line 361) should be corrected.

Response to comment: Thank you again for your careful review and suggestions on our manuscript. We have modified this part of the content in line 414.

Some sentences are overly long; breaking them up would improve readability.

Response to comment: Thanks for your great suggestion on improving the accessibility of our manuscript. We have broken up some long sentences into short ones.

Recommendation: Minor revision before acceptance mainly for clarity, methodological transparency, and narrative tightening.

We have made every effort to thoroughly improve the manuscript. Every piece of revision has been implemented in the revised version; however, these changes do not alter the core content or overall structure of the paper. We sincerely appreciate the editor’s and reviewers’ dedicated efforts and constructive feedback. We hope that the current revision meets the required standards. Once again, we extend our gratitude for your valuable comments and suggestions.

Sincerely yours

Huabao Xiong

Jining Medical University, E-mail: xionghbl@163.com

Re: Spectrum02244-25R1 (**IL-17B protects against uropathogenic *E. coli*-induced kidney injury via macrophage infiltration modulation**)

Dear Prof. Huabao Xiong:

Thank you for the privilege of reviewing your work. Below you will find my comments, instructions from the Spectrum editorial office, and the reviewer comments.

Revision Guidelines

Sincerely,
Gregory Wiedman
Editor
Microbiology Spectrum

Reviewer #1 (Comments for the Author):

This is a resubmission of the manuscript by Wang et al., describing the role of IL17B signaling in immunity against urinary tract infection. In most parts, the authors address the concerns raised in the previous review. Yet, I still have several concerns that must be addressed.

Lines 128-130. Correct the statement to say "monocytes" instead of "macrophages". There are no macrophages in the blood. Also, the sentence is duplicated.

Line 147. No need for a new paragraph. The section should be combined with the previous paragraph since both sections describe histology scoring.

For all qPCR data. The authors must indicate how normalization was performed. This should be mentioned in the figure legends and materials and methods section.

The material and methods section should be expanded. There are insufficient experimental details in each section.

Line 214. Explain what is shown in Figure 1A relates to IL17A, and it is not clear what it means to be "involved." Maybe you should state that IL17A is induced during infection, and the lack of IL17A signaling increases susceptibility to UTI.

Line 233. Insert a space between the word "published" and the reference.

In Figure 2A, the level of inflammatory monocytes in both WT and IL-17B KO decreases based on the flow cytometry histograms, but no significant changes in the graph. The authors need to select better representative histograms. It also seems that there are differential changes in monocytes and neutrophils in WT vs. IL-17B KO without infection.

In all figures, the authors must mark all statistical analysis comparisons. When you have two variables and two conditions, there should be four comparisons. For example, for Figure 2A, the four comparisons should include WT control vs. WT infected, IL-17B KO control vs. IL-17B KO infected, WT infected vs. IL-17B KO infected, and WT control vs. IL-17B KO control. This is true for many panels, including Figure 1H, Figure 2A-2D, Figure 3A-3D, and Figure 4B.

In Figure 2G, the renal cortex of IL17B KO mice received control liposomes, seems to be less damaged and substantially differs from the image presented in Figure 1I. A similar observation is also present in Figure 3G; the renal cortex images of IL17B KO mice in Figure 3G seem to be less damaged and do not reflect the quantification.

In lines 243-245, the statement "Compared to infected WT mice, IL-17B^{-/-} mice exhibited increased neutrophil percentages in the peripheral blood, spleen, and kidney (Figure 3A-3C)" is not supported by the graphs. Figure 3A and B did not show significant changes between infected and uninfected mice in both WT and IL17B KO. Also, Figure 2C showed no significant neutrophil migration into the kidneys of infected WT and IL17B KO mice. The authors must correct the interpretation of the results.

The efficiency of monocyte and neutrophil depletion should be included in Figures 2D and 3G, respectively.

Line 252-254 Although there is a trend of reduced kidney and bladder damage upon neutrophil depletion, the graph did not show statistical significance. The authors must fix the interpretation of the results to match the results.

Lines 262-263. The authors stated, "IL-17B^{-/-} mice treated with rIL-17B showed significantly reduced renal injury (Figure 4A and 4B). Yet the graphs did not show a statistically significant result.

In Figure 4, what does "NS" stand for? A proper control for rIL-17B should be performed. What is the source of rIL-17B? Usually, recombinant proteins are reconstituted with BSA to increase stability. If BSA was used, then a solution with BSA should be used as a control.

Statistical analysis for Figure S1 panels was incorrectly performed.

Line 285-286. The results interpretation by the authors must be fixed. The authors stated that "qPCR and ELISA results showed elevated levels of IL-1 β , IL-6, and IL-12/23 p40 in IL-17B^{-/-} mice (Figure 5A and 5B)". However, there is no significant difference between infected WT and IL-17B KO. There seems to be a trend of increased cytokine levels. The protein levels showed only IL-6 and maybe IL-12, but the graph bars are very small. IL-12 cytokines should be graphed separately since there is a low level produced when compared to other cytokines.

For Figure 5C, it is not clear which chemokine changes significantly between the conditions. The authors must provide a supplemental table showing all conditions and the p-values between each comparison. In addition, proper control for rIL-17B should be performed.

What does "(53)" represent in lines 662 and 678?

Figure legends. The authors should refer to each panel separately. It is very confusing to tell what each panel represents.

Lastly, using in vitro macrophage culture, the authors claim IL-17B attenuated the expression of macrophage CCL2, CCL3, and CCL7 (Lines 294-296). First, macrophage differentiation was performed using GM-CSF. This is not the classical differentiation

cytokine used to generate macrophages. Differentiating bone marrow cells with GM-CSF will generate dendritic-like cells. The authors must repeat these experiments using M-CSF. Also, the authors should test whether expression of CCL-2, -3, and -7 is driven by IL-17B signaling by testing their expression levels in infected IL-17B KO macrophages.

Reviewer #2 (Comments for the Author):

The authors investigate the role of IL-17B (and its receptor) in kidney damage associated with urinary tract infection (UTI) caused by Uropathogenic Escherichia coli (UPEC). Using a mouse model (wild-type vs IL-17B-knockout), they show that IL-17B deficiency leads to greater mortality, increased bacterial colonization in bladder and kidney, worsened kidney histopathology, and elevated plasma creatinine after UPEC infection. They observe increased infiltration of M1-type macrophages (and neutrophils) in kidneys of IL-17B^{-/-} mice. In contrast, administration of recombinant IL-17B (rIL-17B) reduces immune cell infiltration, lowers bacterial burden, and alleviates kidney injury. Mechanistically, they propose that IL-17B limits expression of chemokines (CCL2, CCL3, CCL7) that recruit macrophages, thus modulating macrophage/neutrophil infiltration and preventing excessive inflammation. The authors conclude IL-17B serves a protective, immunomodulatory role in UPEC-induced kidney injury, and may represent a potential adjunct therapeutic target for severe UTIs.

January 18, 2026

Dear Editor and Reviewers:

We would like to extend our true thanks for your continued support and kind attention to our manuscript entitled "IL-17B Protects Against Uropathogenic *E. coli*-Induced Kidney Injury via Modulation of Macrophage Infiltration" (Submission ID: Spectrum02244-25R1). Throughout the revision process, we considered each suggestion carefully. Those helped us to improve the quality of our manuscripts. The revised version has now been uploaded. Meanwhile, a detailed response to your feedback is included below.

Reviewer #1:

This is a resubmission of the manuscript by Wang et al., describing the role of IL17B signaling in immunity against urinary tract infection. In most parts, the authors address the concerns raised in the previous review. Yet, I still have several concerns that must be addressed.

Lines 128-130. Correct the statement to say "monocytes" instead of "macrophages". There are no macrophages in the blood. Also, the sentence is duplicated.

Response to comment: We are truly grateful for your careful and thoughtful review, and we apologize that our initial consideration did not fully account for all aspects. We changed "macrophages" to "phagocytes", which was shown in lines 127-129.

Line 147. No need for a new paragraph. The section should be combined with the previous paragraph since both sections describe histology scoring.

Response to comment: We are very grateful for your detailed comments and suggestions. We have combined these two paragraphs into one. This was shown from lines 137-152.

For all qPCR data. The authors must indicate how normalization was performed. This should be mentioned in the figure legends and materials and methods section.

The material and methods section should be expanded. There are insufficient experimental details in each section.

Response to comment: Thank you again for your careful review and suggestions on our manuscript. We have added more details to this section and expanded it to provide a more comprehensive description, which was shown in lines 174-182.

Line 214. Explain what is shown in Figure 1A relates to IL17A, and it is not clear what it means to be "involved." Maybe you should state that IL17A is induced during infection, and the lack of IL17A signaling increases susceptibility to UTI.

Response to comment: Thank you for your thorough review and constructive suggestions. We have changed this part to "previous studies have established that the expression of IL-17A increased in UPEC-infected bladders and kidneys. When IL-17A is deficient, the colonization of UPEC in the kidneys and bladders of mice

significantly increases. This study also showed that the expression of IL-17A in the kidneys of mice has significantly increased" and provided a description of the specific function of IL-17A.

Line 233. Insert a space between the word "published" and the reference.

Response to comment: We are very grateful for your detailed comments. We have added spaces. This was shown in line 237.

In Figure 2A, the level of inflammatory monocytes in both WT and IL-17B KO decreases based on the flow cytometry histograms, but no significant changes in the graph. The authors need to select better representative histograms. It also seems that there are differential changes in monocytes and neutrophils in WT vs. IL-17B KO without infection.

Response to comment: We are very grateful for the detailed review. We apologize for the diagram we used. We have corrected all the flow cytometry representative graphs, and replaced the representative graph. Those were shown in figures 2A-2C and 3A-3C.

In all figures, the authors must mark all statistical analysis comparisons. When you have two variables and two conditions, there should be four comparisons. For example, for Figure 2A, the four comparisons should include WT control vs. WT infected, IL-17 KO control Vs. IL-17B KO infected, WT infected Vs. IL-17B KO infected, and WT control Vs. IL-17 KO control. This is true for many panels, including Figure 1H, Figure 2A-2D, Figure 3A-3D, and Figure 4B.

Response to comment: Thank you for your thorough review and constructive suggestions. We have added the analysis. However, 1H and 4B show the kidneys and bladder placed in the same graph. On the left is the score for kidney tissue damage, and on the right is the score for bladder tissue damage. So, we did not re-analyze these two graphs.

In Figure 2G, the renal cortex of IL17B KO mice received control liposomes, seems to be less damaged and substantially differs from the image presented in Figure 1I. A similar observation is also present in Figure 3G; the renal cortex images of IL17B KO mice in Figure 3G seem to be less damaged and do not reflect the quantification.

Response to comment: We are very grateful for the detailed review. UPEC infection mainly causes pyelonephritis during the acute stage, causing damage to the renal papillae and renal medulla of the kidneys, and rarely to the renal cortex. This is also reflected and explained in the renal score [1-2].

[1] Choudhry N, Li K, Zhang T, Wu K Y, Song Y, Farrar CA, Wang N, Liu C F, 565 Peng Q, Wu W, Sacks SH, Zhou W. 2016. The complement factor 5a receptor 566 1 has a pathogenic role in chronic inflammation and renal fibrosis in a murine 567 model of chronic pyelonephritis. *Kidney International* 90:540 554. 568.

[2] Li K, Wu KY, Wu W, Wang N, Zhang T, Choudhry N, Song Y, Farrar CA, Ma 569 L, Wei L I, Duan ZY, Dong X, Liu EQ, Li ZF, Sacks SH, Zhou W. 2017. 570 C5aR1

promotes acute pyelonephritis induced by uropathogenic *E. coli*. JCI 571 Insight 2.

In lines 243-245, the statement "Compared to infected WT mice, IL-17B^{-/-} mice exhibited increased neutrophil percentages in the peripheral blood, spleen, and kidney (Figure 3A-3C)" is not supported by the graphs. Figure 3A and B did not show significant changes between infected and uninfected mice in both WT and IL17B KO. Also, Figure 2C showed no significant neutrophil migration into the kidneys of infected WT and IL17B KO mice. The authors must correct the interpretation of the results.

Response to comment: We are grateful for the valuable suggestions. We have made revisions to this part of the content, separating the description of the kidneys from that of the peripheral blood and the spleen. For more details, please refer to lines 249-251.

The efficiency of monocyte and neutrophil depletion should be included in Figures 2D and 3G, respectively.

Response to comment: We appreciate your valuable suggestions. Since the figures 2 and 3 are a bit big, we placed the exhaustion efficiency in Figure S1 and Figure S2G-S2I.

Line 252-254 Although there is a trend of reduced kidney and bladder damage upon neutrophil depletion, the graph did not show statistical significance. The authors must fix the interpretation of the results to match the results.

Response to comment: Thank you for your valuable professional opinion. We have made modifications to similar parts in this manuscript. The details were shown in the article. In this case, we added the word "trend" to this sentence, as can be seen in lines 260-261.

Lines 262-263. The authors stated, "IL-17B^{-/-} mice treated with rIL-17B showed significantly reduced renal injury (Figure 4A and 4B). Yet the graphs did not show a statistically significant result.

Response to comment: We are grateful for the valuable suggestions. Here we only described the damage to the kidneys, but did not mention the damage to the bladder. There are significant differences in the damage to the kidneys, and the description of the bladder has now been added. Please refer to line 271-272 for details.

In Figure 4, what does "NS" stand for? A proper control for rIL-17B should be performed. What is the source of rIL-17B? Usually, recombinant proteins are reconstituted with BSA to increase stability. If BSA was used, then a solution with BSA should be used as a control.

Response to comment: Dear reviewer, thank you for the professional comments. Due to the short treatment time for the mice, and the cytokines were always stored on ice during the operation, a dilution with normal saline was used. The control group was also injected with the same amount of normal saline to ensure the consistency of the experimental conditions.

Statistical analysis for Figure S1 panels was incorrectly performed.

Response to comment: We are very grateful to the expert for the valuable suggestions. We sincerely apologize for the incorrect annotation. Figure S1 has been reanalyzed using two-way ANOVA, and we have now included the comparative analysis between WT blank and IL-17B^{-/-} balnk. The updated results are presented in the figure S2.

Line 285-286. The results interpretation by the authors must be fixed. The authors stated that "qPCR and ELISA results showed elevated levels of IL-1 β , IL-6, and IL-12/23 p40 in IL-17B^{-/-} mice (Figure 5A and 5B)". However, there is no significant difference between infected WT and IL-17B KO. There seems to be a trend of increased cytokine levels. The protein levels showed only IL-6 and maybe IL-12, but the graph bars are very small. IL-12 cytokines should be graphed separately since there is a low level produced when compared to other cytokines.

Response to comment: We appreciate your valuable suggestions. We have split this picture and revised the corresponding textual descriptions accordingly.

For Figure 5C, it is not clear which chemokine changes significantly between the conditions. The authors must provide a supplemental table showing all conditions and the p-values between each comparison. In addition, proper control for rIL-17B should be performed.

Response to comment: Thank you for the valuable suggestions from the experts. In the supplementary materials, we have separately presented all the chemokines. The control for rIL-17B was the untreated wild-type mice of CFT073. We believe this can demonstrate that rIL-17B can serve as a therapeutic approach under normal conditions. Those were shown in Figure S4

What does "(53)" represent in lines 662 and 678?

Response to comment: We are very grateful for your meticulous review, and deeply sorry for the mistake we made. We have removed the cited literature. The modified content is shown in lines 670 and 691.

Figure legends. The authors should refer to each panel separately. It is very confusing to tell what each panel represents.

Response to comment: We sincerely appreciate your valuable and professional suggestions. We have made modifications to these figure legends. For details, please refer to the legends themselves.

Lastly, using in vitro macrophage culture, the authors claim IL-17B attenuated the expression of macrophage CCL2, CCL3, and CCL7 (Lines 294-296). First, macrophage differentiation was performed using GM-CSF. This is not the classical differentiation cytokine used to generate macrophages. Differentiating bone marrow cells with GM-CSF will generate dendritic-like cells. The authors must repeat these experiments using M-CSF. Also, the authors should test whether expression of CCL-2, -3, and -7 is driven by IL-17B signaling by testing their expression levels in infected IL-17B KO macrophages.

We have already changed this part to a re-test using macrophages differentiated from M-CSF, which were shown in Figure 5E-5L.

Reviewer #2 (Comments for the Author):

The authors investigate the role of IL-17B (and its receptor) in kidney damage associated with urinary tract infection (UTI) caused by Uropathogenic Escherichia coli (UPEC). Using a mouse model (wild-type vs IL-17B-knockout), they show that IL-17B deficiency leads to greater mortality, increased bacterial colonization in bladder and kidney, worsened kidney histopathology, and elevated plasma creatinine after UPEC infection. They observe increased infiltration of M1-type macrophages (and neutrophils) in kidneys of IL-17B^{-/-} mice. In contrast, administration of recombinant IL-17B (rIL-17B) reduces immune cell infiltration, lowers bacterial burden, and alleviates kidney injury. Mechanistically, they propose that IL-17B limits expression of chemokines (CCL2, CCL3, CCL7) that recruit macrophages, thus modulating macrophage/neutrophil infiltration and preventing excessive inflammation. The authors conclude IL-17B serves a protective, immunomodulatory role in UPEC-induced kidney injury, and may represent a potential adjunct therapeutic target for severe UTIs.

We are truly grateful for the thoughtful guidance provided by the reviewers. According to the revision suggestions, we added Supplementary Figures 1 and 4. In Supplementary Figure 2, we added figures H-I. In Figure 5, some contents were replaced. These modifications do not affect the core content or overall structure of the paper. Your insightful comments have not only greatly improved the quality of our manuscript but have also been a valuable learning experience that has enriched my research abilities. We sincerely appreciate the editor and reviewers for your meticulous and responsible review as well as your constructive suggestions, and would like to extend our heartfelt thanks once again. We look forward to hearing your good news.

Sincerely yours

Huabao Xiong

Jining Medical University, E-mail: xionghbl@163.com

Re: Spectrum02244-25R2 (IL-17B protects against uropathogenic *E. coli*-induced kidney injury via macrophage infiltration modulation)

Dear Prof. Huabao Xiong:

Thank you for the privilege of reviewing your work. Below you will find my comments, instructions from the Spectrum editorial office, and the reviewer comments.

Revision Guidelines

Sincerely,
Gregory Wiedman
Editor
Microbiology Spectrum

Reviewer #1 (Comments for the Author):

The authors addressed all my concerns. However, there are still minor corrections that must be made.

1. The statement in lines 218-219 should caption Figure 1B, not Figure 1A.
2. The authors should comment on why there were higher monocytes in clodronate liposome-injected mice (Figure S1A).
3. There is still a persistent use of the wrong test to do statistical analysis. Figure 1H requires a Two-way Anova and not a

Student's t-test. Indicate in the figure legend what is the test done for Figure 1C graph (mice survival curves). A Student's t-test should be used to calculate significance in the Supplement 1 graphs.

February 27, 2026

Dear Editor and Reviewer:

We sincerely appreciate your continued support and thoughtful consideration of our manuscript entitled “IL-17B Protects Against Uropathogenic *E. coli*-Induced Kidney Injury via Modulation of Macrophage Infiltration” (Submission ID: Spectrum02244-25R3). In this revision process, we have made careful revisions to every piece of the suggestions received from the reviewer. These insights significantly strengthened the scientific rigor and clarity of the manuscript. The revised manuscript has now been submitted through the online system. A point-by-point response to all comments is provided below.

Reviewer #1 (Comments for the Author):

The authors addressed all my concerns. However, there are still minor corrections that must be made.

1. The statement in lines 218-219 should caption Figure 1B, not Figure 1A.

Response to comment: We are truly grateful for your careful and thoughtful review, and we apologize that our initial consideration did not fully account for all aspects. We changed "1A" to "1B", which was shown in line 219.

2. The authors should comment on why there were higher monocytes in clodronate liposome-injected mice (Figure S1A).

Response to comment: Dear Reviewer, we sincerely thank you for your exceptionally thorough and insightful review. We share your initial concern and have carefully re-examined the data. We found that in wild-type (WT) mice treated with Clodronate liposomes, the number of circulating inflammatory monocytes was significantly reduced (Fig. S1B). In contrast, in IL-17B^{-/-} mice, macrophage depletion increased peripheral inflammatory monocytes (Fig. S1A). Prior studies have demonstrated that IL-17B is related to autophagy. Thus, we hypothesize that IL-17B deficiency may impair monocyte phagocytic capacity, while its impact on macrophage phagocytosis is slight. We have described this phenomenon in lines 365-371.

3. There is still a persistent use of the wrong test to do statistical analysis. Figure 1H requires a Two-way Anova and not a Student's t-test. Indicate in the figure legend

what is the test done for Figure 1C graph (mice survival curves). A Student's t-test should be used to calculate significance in the Supplement 1 graphs.

Response to comment: We appreciate your valuable suggestions and sincerely apologize for the annotation errors identified in this manuscript. We performed histopathological scoring on kidneys and bladders from blank control mice. Then, we used two-way ANOVA to analyze. Meanwhile, we have annotated the analytical method used in the legends of Fig.1 and Fig. 1S.

We sincerely thank the reviewer for the thoughtful and constructive suggestions. These suggestions have enhanced the quality of our manuscript and improved its completeness. All suggested revisions have been carefully implemented. Also, due to lots of genes in mice that are partially homologous to the *ccl14* and *ccl15* genes, we have removed the two genes from Fig. 5C and Fig.4S. These changes refine the exposition without altering the core findings and interpretations. We also gratefully acknowledge the editor and reviewers for their time, expertise, and diligent oversight throughout the review process. We hope hear good news from you.

Sincerely yours

Huabao Xiong

Jining Medical University, E-mail: xionghbl@163.com

Re: Spectrum02244-25R3 (IL-17B protects against uropathogenic *E. coli*-induced kidney injury via macrophage infiltration modulation)

Dear Prof. Huabao Xiong:

Your manuscript has been accepted, and I am forwarding it to the ASM production staff for publication. Your paper will first be checked to make sure all elements meet the technical requirements. ASM staff will contact you if anything needs to be revised before copyediting and production can begin. Otherwise, you will be notified when your proofs are ready to be viewed.

Sincerely,
Gregory Wiedman
Editor
Microbiology Spectrum